

# The marine reservoir age of Greenland coastal waters

Christof Pearce[1], Karen Søby Özdemir[1], Ronja Forchhammer[1], Henrieka Detlef[1], Jesper Olsen[2]

[1]Department of Geoscience, Arctic Research Centre and iClimate, Aarhus University, Høegh Guldbergs Gade 2, 8000 Aarhus, Denmark

[2]Aarhus AMS Centre, Department of Physics and Astronomy, Aarhus University, 8000 Aarhus C, Denmark

*Correspondence to*: Christof Pearce (christof.pearce@geo.au.dk)

**Abstract.**

Knowledge of the marine reservoir age is fundamental for creating reliable chronologies of marine sediment archives based on radiocarbon dating. This age difference between the $^{14}$C age of a marine sample and that of its contemporaneous atmosphere

is dependent on several factors, among others ocean circulation, water mass distribution, terrestrial runoff, upwelling, sea-ice cover and is therefore spatially heterogenous. Anthropogenic influence on the global isotopic carbon system, mostly through atmospheric nuclear tests, has complicated the determination of the regional reservoir age correction ΔR, which therefore can only be measured on historic samples of known age. In this study we expand on the few existing measurements of ΔR for the coastal waters around Greenland, by adding 92 new radiocarbon dates on mollusks from museum collections. All studied

mollusk samples were collected during historic expeditions of the late 19$^{th}$ and early 20$^{th}$ centuries and besides coastal sites around Greenland, the dataset also includes localities from the western Labrador Sea, Baffin Bay, and the Iceland Sea. Together with existing measurements, the new results are used to calculate average ΔR values for different regions around Greenland, all in relation to Marine20, the most recent radiocarbon calibration curve. To support further discussions and comparison with previous datasets, we introduce the term ΔR$_{13}$ where the suffix 13 refers to the previous calibration curve Marine13. Our study

explores the links between the marine reservoir age and oceanography, sea ice cover, water depth, mollusk feeding habits, and the presence of carbonate bedrock. Although we provide regional averages, we encourage people to consult the full catalogue of measurements and determine a suitable ΔR for each case individually, based on the exact location including water depth. Despite this significant expansion of the regional reservoir age database around Greenland, data from the northern coast, directly bordering the Arctic Ocean remains missing.

## 1 Introduction

### 1.1 Radiocarbon dating of marine samples

The most common method for obtaining ages of sediments younger than 50 thousand years is radiocarbon dating. This method relies on the uptake of radiocarbon ($^{14}$C) by plants and animals in equilibrium with their environment and, following death of the organism, the subsequent disconnection between them. The $^{14}$C in the sample then decays to $^{14}$N through beta emission

with a half-life of 5730 years. Provided the carbon within the sample has remained a closed system, and the initial $^{14}$C activity





is known, the remaining $^{14}$C concentration of the sample can be used to calculate its age. To account for the temporal variations in $^{14}$C concentration in the atmosphere and oceans, radiocarbon dating relies on calibration data, which enables the conversion to calibrated ages (Heaton et al., 2021). The most recent such calibration datasets are IntCal20 (Reimer et al., 2020), SHCal20 (Hogg et al., 2020), and Marine20 (Heaton et al., 2020), for northern hemisphere atmospheric, southern hemisphere

atmospheric, and marine samples, respectively. For the purpose of radiocarbon dating, the atmosphere is considered to be well-mixed, with a uniform global distribution of $^{14}$C content in each hemisphere. In contrast, the ocean is a more heterogeneous environment due to spatially varying stratification and upwelling intensity and the overall relatively slow mixing of water masses. The intermediate and deep ocean are typically much more depleted in $^{14}$C compared to the surface ocean, which can lead to differences in radiocarbon age of centuries to millennia across the water column (Broecker et al., 1960; Matsumoto,

2007). The age of the surface ocean is thus influenced by different carbon reservoirs each with their own $^{14}$C content, a phenomenon called the reservoir effect (Stuiver and Braziunas, 1993). The reservoir age is time-dependent, termed R(t), and is defined as the difference between the radiocarbon age of a sample from within the reservoir and the age of the contemporaneous atmosphere (Stuiver et al., 1986). In the Marine20 calibration dataset, the modelled global average marine reservoir age is approximately 550 years (Heaton et al., 2020), but this value varies spatially due to differences in ocean

circulation, upwelling, runoff, and gas exchange with the atmosphere. When calibrating marine radiocarbon dates, this spatial heterogeneity is accounted for by applying a regional reservoir age correction, ΔR (Reimer and Reimer, 2001; Stuiver et al., 1986), which is defined as the contemporaneous difference between the 14C age of the reservoir and the global marine calibration curve, e.g. Marine20 or Marine13.

In principle, ΔR could be calculated from the radiocarbon age of a modern marine sample, but anthropogenic contamination

has complicated this approach on recent samples (Mangerud, 1972). Increased input of $^{14}$C from bomb testing (Hesshaimer et al., 1994), and, to a lesser extent, injection of $^{14}$C-depleted $CO_2$ from the burning of fossil fuels (Suess, 1955), have made it impossible to measure the modern reservoir age of the surface ocean. To determine ΔR, we therefore rely on samples that predate the contamination (ca. 1950 CE), and where the age of the sample is known or independently determined through either tephrochronology (Pearce et al., 2017; Austin et al., 1995; Olsen et al., 2014), or paired marine/terrestrial dating

(Ascough et al., 2005). The most common approach based on recent samples of known age is the use of museum collections where typically the exact locality, date and other collection details are available and reliable. Several hundred different studies were made to study the local reservoir age, and an overview of all shallow water (<75m) samples is provided online at The Marine Reservoir Database (Reimer and Reimer, 2001). This database is continuously updated and maintained, and currently more than 1000 data entries for the regional reservoir correction are listed, together with relevant metadata. The coverage is

global, but not evenly distributed geographically, and some regions (e.g. Western Africa, North Siberia, parts of Greenland) are represented by only very few sample stations (Alves et al., 2018; Reimer and Reimer, 2001).



## 1.2 The Marine20 calibration curve

The publication of the most recent marine radiocarbon calibration curve Marine20 (Heaton et al., 2020), introduced a new global average for the marine reservoir age compared to previous such datasets, including its predecessor Marine13 (Reimer et al., 2013). The globally averaged value of the reservoir age in the Marine20 model is significantly higher (about 155 years) compared to Marine13, which made it necessary to recalculate $\Delta R$ values so they could be used with the new Marine20 curve (Heaton et al., 2020). This has led to some confusion, however, since the paleoceanography community had gotten used to referring to regional reservoir correction values relative to the Marine13 or previous calibration curves (see also discussion in Pieńkowski et al., 2022). Calibration of a radiocarbon date using the Marine20 curve but corrected by an old (pre-2020) $\Delta R$ value, simply results in the wrong final calibrated age so it is crucial to use the appropriate updated $\Delta R$ value for the new curve. Examples of circum-Greenland studies where Marine20 was used, while the ages were corrected by an outdated $\Delta R$ value are plenty (Olsen et al., 2022; Devendra et al., 2022; Sha et al., 2022; Peral et al., 2022; Allaart et al., 2021) and this has unfortunately lead to errors in the final age models. To simplify the discussion and to enable us to cite previously used regional reservoir age correction values, we introduce the term $\Delta R_{xx}$ where xx stands for the publication year of calibration curve. If no suffix is included, the term $\Delta R$ is here meant relative to the newest calibration curve Marine20. As an example, in Disko Bay, a regional reservoir correction of $140 \pm 35$ $^{14}$C years based on measurements of pre-bomb *Astarte* mollusks (McNeely et al., 2006; Lloyd et al., 2011) has been used in multiple paleoceanographical studies (Moros et al., 2016; Krawczyk et al., 2017; Li et al., 2017; Ouellet-Bernier et al., 2014). From here onwards, we would cite such values as $\Delta R_{13} = 140 \pm 35$ $^{14}$C years, here referring to the Marine13 curve. In this case of Disko Bay, the new correction value relative to Marine20 based on the same two mollusk samples is $\Delta R = 20 \pm 76$ $^{14}$C years (Reimer and Reimer, 2001). Thus, although the total reservoir age R has not changed, the regional reservoir correction $\Delta R$ is different because of the different calibration curves used.

## 1.3 The Marine20 curve in polar regions

The most recent marine calibration curve update (Marine20) made it explicit that the calibration curve does not apply for calibration of samples from polar regions (Heaton et al., 2020). This is of course, a particular challenge for studies focusing on Greenland. The main reason to avoid Marine20 outside latitudes from 40° S – 40° N is the variability of air-sea gas exchange due to changes in sea-ice cover in these cold regions. Especially in glacial periods, extensive sea-ice cover would reduce ventilation and can lead to significant increases in the local reservoir age of over 1000 years compared to modern values (Heaton et al., 2020). This limitation of using radiocarbon dating of marine samples from high latitudes is however not unique to the latest Marine20 dataset; it has been true for previous calibration curves as well, although not as explicitly mentioned. Simple box models have shown a consistent link between sea-ice cover and the radiocarbon reservoir age (Bard et al., 1994), and high-resolution studies using tephrochronology coupled with radiocarbon dates have documented North Atlantic reservoir ages of > 2000 years during deep stadials of MIS3 (Olsen et al., 2014). Despite the advice against using Marine20 in polar regions, it has already been used extensively for this purpose since its publication (Pados-Dibattista et al., 2022; Glueder et al.,



2022; Hansen et al., 2022; El bani Altuna et al., 2021; Brouard et al., 2021; Stevenard et al., 2022; Davies et al., 2022; Jackson
et al., 2022). Since publication of Marine20, a more recent paper by the same main authors has now been published and clarifies
this issue. Heaton et al. (2023) state that their warning about using the marine calibration curve in polar regions also applies to
previous versions and is not something new and unique to Marine20. The latest advice is now to use, with caution, the Marine20
calibration curve for polar samples of Holocene age (interglacial, non-$^{14}$C-depleted surface ocean), and to use it as a lower
bound for calibration samples from glacial periods during which the surface oceans may be more depleted (Heaton et al.,
2023). This additional challenge with calibration of polar marine radiocarbon dates, emphasizes the importance of studies such
as the one presented here, aiming to expand the knowledge of spatial variability of ΔR in these regions.

### 1.4 The marine reservoir age around Greenland

As anywhere else in the marine realm, the regional radiocarbon reservoir age around Greenland is influenced by ocean currents,
terrestrial runoff, mixing of water masses of different ages, and variable exchange between the surface ocean and atmosphere.
In the Arctic, the latter is heavily influenced by the presence of sea ice, which may inhibit mixing between surface waters and
the overlying air. Greenland has perennial sea-ice cover along its northern coastline while the southern coast typically
experiences year-round ice-free conditions. These contrasting sea-ice conditions create a pattern of older surface waters along
the Arctic Ocean and younger waters at the lower latitudes coastal waters facing the North Atlantic Ocean. In the surface
ocean, East Greenland is characterized by Polar Water flowing out from the Arctic Ocean through the Fram Strait with the
East Greenland Current. Similarly, Northwest Greenland receives Polar Water coming through the Nares Strait and the
Canadian Arctic Archipelago, also containing a larger share of older Pacific Water, originating from the Bering Strait (Jones
et al., 2003). This pattern is further complicated, however, by the presence of well-ventilated, and thus younger, Atlantic waters
that occupy the subsurface waters. Atlantic water is known to reach the Greenlandic coast in many locations, typically driven
by the bathymetry, e.g. where deep glacially-formed cross-shelf troughs allow this subsurface water connection (Schaffer et
al., 2020; Sutherland et al., 2013; Millan et al., 2018). The perceived reservoir age correction ΔR as measured on certain
organisms may also be impacted by the presence of carbonate bedrock in the area. This is of particular influence on deposit
feeders which may take up more old dissolved carbonate than filter feeding mollusks (Mangerud et al., 2006; England et al.,
2013; Forman and Polyak, 1997).

Existing measurements of regional marine reservoir age ΔR values from coastal waters of Greenland are relatively sparse and
highly clustered around a few localities such as Disko Bay and Scoresby Sound (Hjort, 1973; McNeely et al., 2006; Tauber
and Funder, 1975). In the north, no data at all exists above latitudes of 77° and 79° N, in East and West Greenland, respectively
(Reimer and Reimer, 2001). In the south, between 68° N and down to Greenland's southernmost point at Cape Farewell, only
4 measurements exist and are almost exclusively on marine mammals (polar bears and humans) for which the feeding area and
marine fraction of their diet is not exactly known (Olsson, 1980). This heterogeneously spaced dataset has allowed different
studies to use various values for the regional reservoir correction. In studies of Holocene paleoceanography offshore





Greenland, the most commonly used value for the reservoir age correction (prior to publication of Marine20), $\Delta R = 0$ $^{14}$C years, depending on the year of publication either as $\Delta R_{04}$, $\Delta R_{09}$, or $\Delta R_{13}$ (Knudsen et al., 2008; Levac et al., 2001; Jensen et al., 2004; Lassen et al., 2004; Nørgaard-Pedersen and Mikkelsen, 2009; Andresen et al., 2011; Seidenkrantz et al., 2008; Lloyd,

2006; Moros et al., 2006; Møller et al., 2006; Andrews and Jennings, 2014; Andresen et al., 2013). In Disko Bay, West Greenland, Lloyd et al. (2011) argued for $\Delta R_{09} = 140 \pm 30$ $^{14}$C years which was subsequently used in following studies close to the study site (Moros et al., 2016; Krawczyk et al., 2017; Perner et al., 2013; Ouellet-Bernier et al., 2014; Li et al., 2017) as well as the wider region all the way to Sissimut in the south (Erbs-Hansen et al., 2013) and Upernavik in the north (Hansen et al., 2020). In East Greenland, some studies have differentiated between sites north and south of the Denmark Strait. Jennings

et al. (2011) used $\Delta R_{09} = 0$ $^{14}$C years on the shelf south of Denmark Strait based on the occurrence of several tephra layers, and $\Delta R_{09} = 149 \pm 99$ $^{14}$C years north of Denmark Strait, arguing for a stronger influence of older Polar Water at higher latitudes. A recurring argument for using a $\Delta R$ value of 0 $^{14}$C years is the presence of young Atlantic waters in the subsurface corresponding to the sediment coring location (Andresen et al., 2013), while others have suggested that using the $\Delta R$ value of 0 $^{14}$C years can be considered as a minimum and therefore calibrated ages may be too old (Andrews and Jennings, 2014).

**1.5 Aim of this study**

To allow determination of leads and lags in the climate system over long time scales, there is a need for robust correlations between geological archives from different environments. These correlations between marine, terrestrial, and ice-core records rely on independent, accurate and precise determinations of the age of sediment layers. In the marine environment, one of the main sources of errors in chronology is knowledge of the regional marine reservoir age correction (Alves et al., 2018). The

spatial and temporal heterogeneity of this parameter requires local calibrations, and the present understanding of the modern $\Delta R$ values around Greenland is based on a relatively small number of samples. Besides the low number of available measurements of $\Delta R$, a regional synthesis of the data is lacking for Greenland. Several such investigations exists for neighbouring regions, including Arctic Canada (Coulthard et al., 2010; McNeely et al., 2006; Pieńkowski et al., 2022a) and the northeastern North Atlantic (Mangerud et al., 2006). The aim of this study is to improve the current estimates of the regional

reservoir age around Greenland and to summarize new and existing data into regional recommendations for use with the latest Marine20 calibration curve.

**2 Materials and Methods**

**2.1 Museum sampling**

Mollusc samples were collected from the zoology divisions of the Swedish Museum of Natural History (SMNH) in Stockholm,

Sweden and the Natural History Museum of Denmark (NHMD) in Copenhagen, Denmark. At the SMNH, a combination of a digital database and written catalogues was used to identify potential samples before sampling from the collections. At the NHMD, a lack of such catalogues meant that all samples were selected and sampled directly by visiting the sample storage





facilities and manual inspection of labelled sample containers. In both museums, for selecting suitable mollusc samples, the following criteria were considered:

- Geographic location listed by coordinates (descriptive locations were excluded)
- Location in circum-Greenland waters in the broadest sense (incl. Greenland Sea, North Atlantic Ocean, Labrador Sea, Baffin Bay)
- Known year of sampling and age no younger than 1950 CE
- Sufficient material available so at least 1 intact specimen could be left in the museum collection

A total of 92 samples were included in this study from coastal and shelf waters of Newfoundland, Labrador, Baffin Island, Greenland, and a few open ocean localities in the Labrador Sea and Greenland Sea (Fig. 1). The collection year of the samples ranges between 1865 and 1931 CE, so well before the introduction of bomb-derived $^{14}$C into the global carbon cycle (Hesshaimer et al., 1994). The samples were retrieved during various historic expeditions (Supplementary Table), with most specimens collected during the following 4 sampling campaigns: Ellis Nillson on the Scottish whaling ship *Eclipse* in 1894 to

the northern Baffin Bay, Fredrik von Otter on the *Ingegerd* and *Gladan* Expedition of 1871 to the Labrador Sea and Baffin Bay, Alfred Gabriel Nathorst on the *Antarctica* to East Greenland in 1899, and Eigil Riis-Carstensen on the *Godthåb* Expedition of 1928 to the Labrador Sea and Baffin Bay (Liljequist, 1993; Nathorst, 1900; Riis-Carstensen, 1929; Stolpe, 1894). Most samples (n=75) were taken from the so-called wet collections, i.e. stored in ethanol, but some dry samples (n=17) from the SMNH were included in this study as well. In all wet samples mollusc soft tissue was present before further processing,

ensuring the specimens were alive at the time of sampling. Specimens were transferred to plastic sample containers and transported to Aarhus University for further processing.





Figure 1. Map of Greenland and the northern North Atlantic Ocean, with ocean currents and location of all samples analyzed in this study together with those of the existing ΔR database (Reimer and Reimer, 2001). Areas of deep convection in the Labrador Sea and north of Iceland are colored yellow. LS: Lancaster Sound, NS: Nares Strait. Bathymetry is from GEBCO (GEBCO Bathymetric Compilation Group, 2022).



## 2.2 Laboratory pre-treatment and AMS $^{14}$C dating

Wet samples were placed in a drying oven at 40 °C for several days before any remaining dried-up soft tissue was removed
from the shell. Approximately 10-15 mg of carbonate was removed from the outer, youngest part of the shell for further
treatment. In case of very small mollusc samples, the entire shell was used (smallest sample 5.7 mg). To remove any surface
contamination, the shells were placed in an HCl solution (1M) for 2 hours until 10-20% of the carbonate was dissolved. During
this step, the samples were carefully shaken a few times to ensure the acid could access the entire sample. After acid treatment
the samples were carefully rinsed three times with milliQ water. Each rinsing was done with its own glass and pipette which
were examined after the rinse to ensure that none of the sample was lost in the process. A small amount of water was left to
lower the risk of flushing part of the sample. The samples were dried overnight to remove the excess water and then weighed.
The remaining carbonate was subsequently converted to graphite and $^{14}$C dated using the HVE 1MV tandetron accelerator
AMS system at the Aarhus AMS Centre (Olsen et al., 2017). $^{14}$C dates are reported as uncalibrated $^{14}$C ages BP normalized to
−25‰ according to international convention using online $^{13}$C/$^{12}$C ratios (Stuiver and Polach, 1977).

## 2.3 Calculation of reservoir age per sample

The local reservoir age correction ΔR was calculated for each sample as the difference between the measured radiocarbon age
of the sample and the radiocarbon age of the Marine20 calibration curve (Heaton et al., 2020) corresponding to the year of
collection. To avoid errors and ensure reproducibility, all calculations were performed using the online program *deltar* (Reimer
and Reimer, 2017) with input of the measured radiocarbon age and associated uncertainty, and the year of sample collection,
i.e. the independently known calendar age. The uncertainty of ΔR equals the uncertainty of the radiocarbon date, which
therefore does not take into account the uncertainty of the marine curve (Reimer and Reimer, 2017).

## 2.4 Geographical zonation

To summarize the results and provide broad, regional estimates of the reservoir age corrections around Greenland, the study
area was divided into 7 zones, based on prevailing ocean currents and water masses, see Figure 2. The zone boundaries are
broadly consistent, but not exactly the same as in previous ΔR compilation studies in the area (McNeely et al., 2006; Coulthard
et al., 2010; Pieńkowski et al., 2022a). Zone 1 covers the shelf seas south of Newfoundland, where there is a mixed influence
of Atlantic Water coming from the North Atlantic Current in the south and colder waters from the north via the Labrador
Current (Fig. 1). Zone 2 includes the coastal waters of the western Labrador Sea from northern Newfoundland in the south,
along the Labrador Peninsula, to southern Baffin Island in the north (Fig. 2). These waters are predominantly influenced by
the Labrador Current (Fig. 1), which includes a mixture of cold polar waters coming from the north and Atlantic Water
originating from the West Greenland Current as it branches across the Davis Strait (Fig. 1). Zone 3 covers the northwestern
Baffin Bay and represents a cluster of samples south of Lancaster Sound (Fig. 2). It is characterized by outflow of Arctic



Waters through the Canadian Arctic Archipelago and Nares Strait (Fig. 1). Zone 4 includes Jones Sound south of Ellesmere Island and Smith Sound at the southern end of Nares Strait (Fig. 2) and, similar to Zone 3, is predominantly influenced by

Artic Water. Zone 5 encompasses the northwestern coast of Greenland, from the Davis Strait in the south to Nares Strait in the north (Fig. 2). It is dominated by waters of the West Greenland Current over the shelf (Fig. 1), while the more coastal inner fjord location will be impacted from outflow from the Greenland Ice Sheet as well. Zone 6 stretches all along the coast and shelf of southern Greenland from. It is boundaries with Zone 7 and Zone 5 are set where the Irminger Current merges with the East Greenland Current, to where the West Greenland Current branches of across the Davis Strait, respectively (Fig. 1). Finally,

Zone 7 covers the coastal waters of northeast Greenland, including several sites inside deep fjord systems (Fig. 2). This region is dominated by the East Greenland Current carrying polar surface waters from the Arctic Ocean through the Fram Strait, underlain by recirculated warmer Atlantic Water at water depths >100m (Fig. 1). Five samples that were analyzed in this study fell outside of the zonation: 3 from deep sites in the Greenland Sea and Norwegian Sea, one from northern Iceland, and one from a deep site in the Labrador Sea (Figure 2).




Figure 2. Map of study area with boundaries of geographic zones and locations of all samples used for calculations (wet collection only). The insert shows zonal average ΔR values, based on all data points except the four that fall outside of the zone boundaries. The box and whiskers charts show the interquartile range (box), the minimum and maximum values (whiskers), and the median value (midline in box). The values shown in red for each zone are the weighted mean values and

uncertainty as described in the methods and also listed in Table 1. The number of datapoints is given for each zone, including the number of new values obtained in this study, listed in brackets.



## 2.5 Regional mean reservoir age correction estimates

To provide an average value and uncertainty for each of the regions, we followed the standard methodology as also applied in the Marine Reservoir Age Database (Reimer and Reimer, 2001) to calculate weighted means while incorporating the uncertainty of the original data. For each zone we calculate the ΔR as the weighted mean value using (Bevington and Robinson, 1969):

$$\Delta R_w = \frac{\sum \Delta R_i / \sigma_i^2}{\sum 1 / \sigma_i^2}$$

(1)

where $\Delta R_i$ and $\sigma_i$ are the mean value and uncertainty of calculated local reservoir age offset. The uncertainty of the weighted mean ΔR values is calculated as:

$$\sigma_w = \sqrt{\frac{1}{\sum 1 / \sigma_i^2}}$$

(2)

Where the subscript w indicates that the uncertainty is calculate using the error each $\Delta R_i$. Further we calculated the standard deviation as:

$$\sigma_{std} = \sqrt{\frac{\frac{1}{n-1}\sum \left(\frac{\Delta R_i - \Delta R_w}{\sigma_i}\right)^2}{\frac{1}{n}\sum \frac{1}{\sigma_i^2}}}$$

(3)

Where n is number of samples and $\Delta R_w$ is the regional weighted mean ΔR value.

No form of outlier analyses was performed and all available data, both previously published and new results were included in the zonal mean values. To account for the larger than statistically expected variability in some of the $\Delta R_w$ values we take the maximum value of $\sigma_w$ or $\sigma_{std}$ as the uncertainty on the $\Delta R_w$ values.

## 3 Results

### 3.1 Wet vs dry museum mollusk collections

All dry samples analysed in this study (n=17) are from the same region along the East Greenland coast between 70° and 75° N (Figure 2). Molluscs stored in ethanol were analysed from the same region, which allows comparison between the two samples sets based on their preservation. Although some radiocarbon dates indicate compatible ΔR values, there is a clear





difference between the two datasets, with the dry samples being significantly older than the wet (Suppl. Fig. 2). At a single location, a dry sample (#83, Suppl. Table 1) returned ΔR = 1141 ± 27 $^{14}$C years, which is > thousand years older than others from the same site (Suppl. Fig.2). There is no certainty that the dry samples were collected alive, and they are therefore

considered unreliable for the purpose of determining the local marine reservoir age. All dry samples are excluded from further discussion in this paper, and all reporting of results from here onwards are of wet-preserved samples only.

## 3.2 Results summary

For the entire study area of coastal Greenland, including the Labrador Sea and Baffin Bay, the newly obtained values for ΔR range from -172 to 546 $^{14}$C years, with a median value of ΔR = -39 $^{14}$C years. The geographical range of the samples reaches

from 46 °N south of Newfoundland to approximately 76 °N and 77 °N, on the eastern and western Greenlandic shelf, respectively (Fig. 1). The majority of new samples are from West Greenland, originating mostly from the 1928 Godthåb Expedition (Riis-Carstensen, 1929). Whereas samples in the online marine reservoir correction database are generally restricted to shallow water depths <75m (Reimer and Reimer, 2001), our new samples represent the broader coastal shelf settings and expand to water depths of typically several hundred meters (Figure 1, Supplementary Table 1). The full depth

scale of the newly dated samples ranges from a black mussel collected on the beach in Cumberland Sound (*Musculus niger*, Sample #5, Supplementary Table 1), to a gastropod collected from 2750 meter water depth (mwd) in the northern central Labrador Sea (*Buccinum abyssorum*, Sample #61, Suppl. Table 1).

## 3.3 Reservoir age offsets per region

The new data for the regional marine reservoir age offsets (ΔR) are here reported for each region, as defined in the methods

section, and illustrated in Figure 2. The results of the 75 newly dated mollusk samples from wet collections are combined with previously existing data from the marine reservoir correction database (Reimer and Reimer, 2001). The results per zone are summarized in Table 1. Although there is overlap between the new ΔR values from the entire study area, there are also clear differences between specific zones. The weighted mean ΔR values for all zones fall between 0 and -100 $^{14}$C years, except for Zones 3 and 4, where the ΔR values are higher. The highest values occur in Zone 3, south of Lancaster Sound (Figure 2), with

a mean ΔR value for this zone of 218 $^{14}$C years. The mean values for Greenland coastal waters from all three zones (East, West, and South) all overlap within the calculated uncertainties (Table 1).



*Table 1. Summary of regional ΔR values. Total samples include existing and new samples from this study (amount listed in*
*parentheses). Range, weighted means, and uncertainties are calculated on the full dataset. Zone numbers refer to Figure 2.*

| Zone # | Region | Total samples | ΔR Range ($^{14}$C years) | Weighted mean ($^{14}$C years) |
|---|---|---|---|---|
| 1 | Newfoundland | 6 (+2) | -158 to 22 | -58 ±73 |
| 2 | W. Labrador Sea | 53 (+4) | -154 to 154 | -12 ±76 |
| 3 | NW Baffin Bay | 11 (+9) | 60 to 546 | 218 ±135 |
| 4 | S. Nares Strait | 14 (+6) | -20 to 186 | 71 ±69 |
| 5 | West Greenland | 43 (+30) | -257 to 116 | -49 ±59 |
| 6 | South Greenland | 14 (+11) | -116 to 13 | -79 ±95 |
| 7 | East Greenland | 22 (+8) | -194 to 49 | -46 ±57 |

Five new samples fall outside the geographical zone boundaries: #65, #66, and #75 from the Norwegian Sea and Greenland
Sea, #64 from the shelf north of Iceland, and #61 from the central Labrador Sea. These 5 are mostly from deeper sites (>1500
mwd), except for Sample #64 which is from 388 mwd (Suppl. Figure 1, Suppl. Table 1). All these deep samples returned
relatively young ages, i.e. ΔR < 0 years, including the deepest sample (#61 from 2750 mwd) which returned one of the lowest
values of ΔR in the entire dataset (ΔR = -149 ± 27 $^{14}$C years).

## 4 Discussion

The new results obtained in this study are overall in good agreement with previously existing measurements of the reservoir
age offsets around Greenland and eastern Canada. There is significant overlap between existing and new measurements, and
therefore this expansion adds to the reliability of regional estimates, but the new data also include many localities for which
previously no information was available. The new measurements mostly come from shallow coastal locations, but also includes
many sites on the shelf with water depths >100 m. This spatial expansion of available data is key for paleoceanographical
studies as they are often based on sediment cores from deeper shelf locations where deglacial – Holocene sediments are dated
using benthic organisms (e.g. mollusks or benthic foraminifera). The discussion presented here includes the relation of ΔR to
water mass, water depth, sea-ice cover, and local variability within small areas, but it is not exhaustive, and the reader is
encouraged to consult the complete tables of results in the supplementary information to investigate specific cases.

### 4.1 Spatial patterns and ocean circulation

The overall results as shown in Figure 2 are consistent with the ocean circulation patterns around Greenland, but there are also
some notable observations. As expected, the lowest ΔR values are found in regions where Atlantic Water prevails: Zone 1
south of Newfoundland and Zone 6 in southern Greenland, along the path of the Irminger / West Greenland Current (Figure
1). The oldest waters or highest ΔR values are found along Baffin Island and in the northernmost Baffin Bay (Figure 2: Zones



3 & 4), a region influenced by older Arctic and Pacific Waters coming through the Canadian Arctic (Coulthard et al., 2010), but also by the presence of carbonate in local bedrock which may further explain older radiocarbon ages (England et al., 2013). For the coastal areas surrounding Baffin Bay, there is overlap with another recent reservoir age compilation study focusing on

the Canadian Arctic (Pieńkowski et al., 2022a). The results are mostly compatible, as both studies show a pattern of old values along the Canadian margin and younger waters along the west Greenland coast. The zonal mean ΔR values are however not exactly the same, since this study adds 45 new radiocarbon measurements for Baffin Bay and both studies also use a slightly different geographical zonation (Pieńkowski et al., 2022a). In between the zones with mostly Atlantic versus Arctic and Pacific water influence, lie Zone 2 and Zone 5 and they show intermediate values, likely the result of mixing between the two

endmembers. This is seen in the high variance of the ΔR values in these zones 2 and 5, which also cover a large latitudinal range (Figure 2). Along East Greenland, remarkably, there is not a major change in local reservoir age across the Denmark Strait, which was hypothesized in other studies to be caused by the influence of the Irminger Current (Jennings et al., 2011). The calculated ΔR values for Zones 6 and 7 (Figure 2), respectively south and north of the Denmark Strait (approx. 67°N), are both well below zero and the calculated mean values are not significantly different (Table 1). There is a larger variance of the

ΔR values in Zone 7, however, but this is likely caused by the larger sample size and by the inclusion of many sites inside fjords in this zone (Fig. 2).

## 4.2 Water depth

By definition, the marine reservoir offset is only valid for the surface ocean, i.e. the upper part of the water column that is well-mixed, with the maximum depth is set to 75 m (Stuiver et al., 1986). This same water depth cut-off value of 75 m is used in

the online marine reservoir correction database at calib.org/marine (Reimer and Reimer, 2017), although some exceptions have been made and data from deeper sites outside the surface mixed layer were included in the database (Lougheed et al., 2013). The publication of the latest marine calibration curve, Marine20, mentions a water depth of 100 m for the surface ocean box in the model, thus slightly extending the depth range for the definition of the surface ocean in reservoir age calculations (Heaton et al., 2020). However, despite this policy of restricting the ΔR database to the upper 75-100 m of the surface ocean, the marine

calibration curves are routinely used in paleoceanography on sediment core samples from well outside of this depth range. In fjords or coastal shelf environments, water depths can easily reach several hundred meters, but planktic foraminifera or other calcifying surface ocean dwellers are often absent. The standard approach here is to resort to dating benthic organisms such as benthic foraminifera or molluscs and apply the local reservoir correction ΔR which is originally intended for the surface ocean only. Even for deep ocean sites where sediment cores are dated using planktic foraminifera one could argue that their habitat

range is not restricted to the upper 75 m of the water column, but are often found below that, up to several hundred meters water depth (Kimoto, 2015). This current study and expansion of the circum-Greenland reservoir age database is therefore not strictly limited to samples from surface waters but includes many samples from deeper sites on the shelf and even a few of the deep ocean (Figure 2). The largest range of ΔR values is found in the surface ocean above approximately 150 m (Figure 3). Below this, the spread is less pronounced and, generally, the ΔR values are closer to zero as one goes down the water column





(Figure 3). When calibrating benthic dates from deeper sites one could therefore consider excluding extreme values obtained from surface ocean samples when making the choice of which reservoir correction to apply. Not all zones include results from deeper sites however, and the decreasing ΔR variability with depth can also partially be attributed to the lower amount of data points available for deeper sites (Figure 3). The few measurements of samples from water depths beyond >1000 meter are from sites in the Greenland Sea, Norwegian Sea, and Labrador Sea (Figure 1). These are all at locations characterized by

convection and formation of North Atlantic Deep Water and Labrador Sea Water (Broecker, 1991; Smethie Jr. et al., 2000), which is the likely explanation for the low ΔR values indicating deep waters younger than the average global surface ocean (Figure 3).

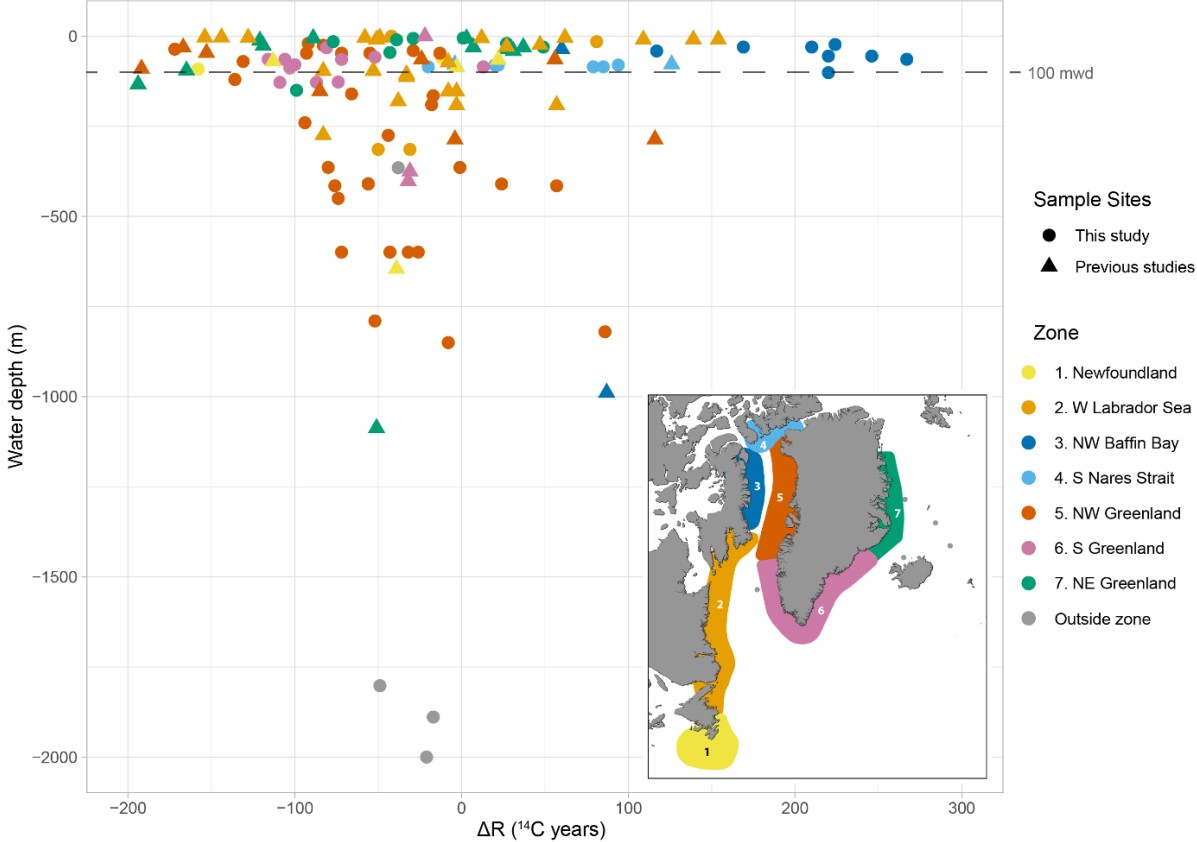

Figure 3. Scatter plot of ΔR values versus water depth of individual sites, colored according to the geographic zone shown in
the insert, as defined in Figure 2. Water depth values for the new sites of this study were obtained from museum catalogues, while the water depth of previous studies was retrieved by finding the GEBCO water depth (GEBCO Bathymetric Compilation Group, 2022) for the coordinates listed in the marine reservoir database of (Reimer and Reimer, 2001). The horizontal and vertical axes are cropped at ΔR = 300 years and 2000 m water depth, respectively, to improve readability of the graph. This resulted in two datapoints falling outside the plotted area, namely #61 (ΔR = -149 ± 27 $^{14}$C yrs, 2750 mwd) and #15 (ΔR = 546
± 25 $^{14}$C yrs, 20 mwd), see also Supplementary Table 1.



### 4.3 Sea ice

Sea ice acts as a physical barrier between the ocean and the atmosphere and prevents the uptake of atmospheric $^{14}CO_2$ in surface waters, thereby increasing the local marine reservoir age. This link has been well-established, and is one of the main

reasons why the latest calibration curve Marine20 is described as not suitable for application in polar regions where sea-ice cover impacts the reservoir age (Heaton et al., 2020). Model experiments suggest a direct relationship between the average annual duration of sea-ice cover and the magnitude of the local reservoir age ΔR (Bard et al., 1994), and sea ice variability in the past has been found to play an important role in ΔR fluctuations on millennial time scales (Butzin et al., 2017). Sea ice conditions around Greenland and the adjacent seas, cover the full range of year-round ice-free conditions in the south to near-

perennial sea-ice cover offshore northeast Greenland, making this study area ideal for investigating the link between sea ice concentration and ΔR. Although there is no direct clear linear relationship between the two variables based on the entire dataset, the highest ΔR values are typically found in areas with elevated sea-ice cover (Figure 4). In sites with annual average sea-ice concentrations of less than 25%, no ΔR values higher than 50 years are found, and where ΔR values exceed 160 years, the sea-ice concentrations are >75% (Figure 4). Within each of the individual geographic zones as defined in Figure 2, there

is also a positive correlation between sea-ice cover and reservoir age, except for Zone 1 around Newfoundland with predominantly ice-free conditions and Zone 4 around Nares Strait (Figure 4). As discussed above, there are other factors influencing the reservoir age and in this specific case, the region with the highest ΔR values (Zone 3, NW Baffin), is not only characterized by elevated sea ice concentrations, but also the influence of Arctic and Pacific waters, and carbonate bedrock, all together contributing to the older waters.








Figure 4. Scatter plot of ΔR values of surface waters (<100 mwd) versus historical sea ice concentrations of individual sites, colored according to the geographic zone shown in the insert, as defined in Figure 2. The dotted lines represent linear trendlines for each of the zones, colored accordingly. Sea ice for each site is the annual average value for the period from 1850 – 1950, derived from Walsh et al. (2017). One datapoint lies outside the plotted area (#15, ΔR = 546 ± 25 $^{14}$C yrs, sea ice 76%), since the horizontal axis is cropped at ΔR = 300 years to improve readability of the graph.

### 4.4 Mollusk feeding habits

A potential obstacle in using marine mollusks as recorders of the $^{14}$C content of the surrounding ocean water is the variability of feeding habits of different species. Depending on the feeding habit, individual species may be taking up carbon from different pools. Suspension feeders, as the name implies, feed on organic matter that is suspended in the water column while deposit feeders take up carbon from material that has settled on the sea floor. This is of great importance in areas with carbonate




bedrock, where sediment pore water may contain old bicarbonate which would offset the age of deposit feeding mollusks. This phenomenon has also been called the '*Portlandia* effect', named after the deposit-feeding mollusk *Portlandia arctica* which was found to be up to several thousand years older than its suspension-feeding counterparts in a study from the Canadian Arctic Archipelago (England et al., 2013). The circum-Greenland dataset analyzed here consists mostly of suspension feeders,

but also numerous deposit feeders. The most common suspension feeders in this study are *Astarte borealis*, *Hiatella arctica*, and *Mytilus edulis*, while the most common deposit feeders are *Macoma calcarea* and *Nuculana permula* (Suppl. Table 1). A direct comparison shows no consistent offset between the two groups, although the deposit feeders do show a tendency towards slightly higher ΔR values (Figure 5). The highest value in the entire dataset is ΔR = 546 ± 25 years, measured on a specimen of *Macoma calcarea* from the northern coast of Bylot Island in Lancaster Sound (Sample #15, Supplementary Fig. 1). This

anomalously high value, combined with the knowledge that it is retrieved from a deposit feeder in a region that is characterized by carbonate bedrock, is thus very likely a result of the *Portlandia* effect, and the date should not be considered reliable. In conclusion, results from the deposit feeders are mostly compatible with those from suspension feeders (Figure 5), but caution should be taken in areas with carbonate bedrock, where the deposit feeders should be avoided.

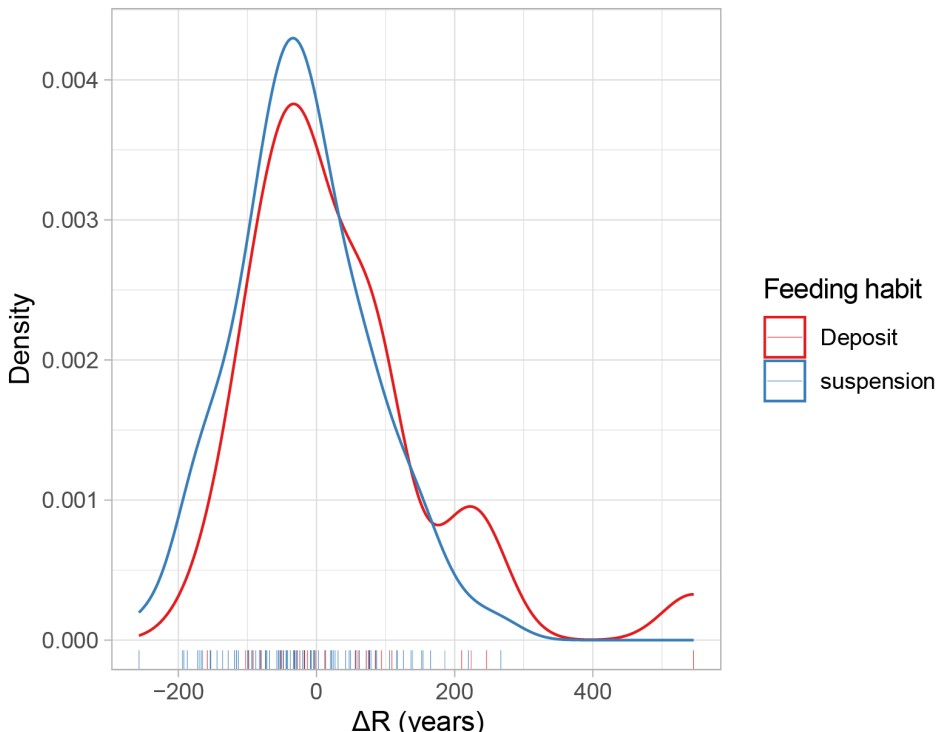

Figure 5. Density plots of ΔR values colored by feeding habit of mollusks. Deposit feeders are plotted in red (n= 27) and suspension feeders in blue (n=98). Information on mollusk feeding habits was obtained from the marine radiocarbon database (Reimer and Reimer, 2001), supplemented by information published on the World Register of Marine Species (WoRMS Editorial Board, 2023).



## 5 Summary and conclusions

This study provides a significant expansion of the available data of the marine reservoir age around Greenland and the adjacent seas. All new data was obtained from radiocarbon measurements on mollusks (n=92) stored in museum collections in Stockholm and Copenhagen. The new data includes many sites from deeper water depths, beyond the surface mixed layer which is traditionally used for the determination of the marine reservoir age. This expansion to deeper water depths, allows the inclusion of samples which are located further offshore, and provides a broader representation of sites that are typically

cored for paleoceanographical studies. When our new data is combined with results from previous studies in the area, the following conclusions can be drawn:

-  Museum sample storage: As the exact age of samples from "dry" collections is possibly unknown, only samples with soft tissue present, stored in "wet" collections, should be used for $\Delta R$ measurements.

-  Water masses and ocean currents: the youngest waters are found in areas dominated by Atlantic Water, while the

oldest are found in regions influenced by Arctic Ocean outflow and influence of Pacific Water.

-  Water depth: The largest variance of $\Delta R$ is found in the surface waters. Further down the water column, the $\Delta R$ values become less variable and are closer to zero. In areas of deepwater formation, even sites beyond 1500 m water depth have $\Delta R$ values $< 0$ $^{14}$C years.

-  Sea ice: On a regional scale, our data suggests a clear link between sea ice cover and radiocarbon age of the underlying

waters. The highest $\Delta R$ values are found in regions with high ($<75\%$) annual sea ice cover, while areas with almost no sea ice cover have typically low ($\Delta R < 50$ $^{14}$C yrs) values.

-  Suspension vs. deposit feeders: In general, $\Delta R$ measurements on mollusks with either feeding habit are compatible. In regions with carbonate bedrock and the presence of old carbon, however, deposit feeders should be avoided, as they are found to overestimate the radiocarbon age of the waters.

For calibration of radiocarbon dates, we provide regional averages as summarized in Table 1, but we encourage users to consult the full table of result provided in the supplementary information. This allows one to choose a $\Delta R$ value based on prevailing water masses, water depth, distance to coast, mollusk species, and other factors, all together providing a value which is as representative of the study site as possible.

Despite this improved estimate of the regional reservoir age around Greenland, these values remain only valid for the modern

situation and most likely large parts of the current Holocene interglacial. Other studies have shown large temporal variability over millennial timescales linked to major changes in ocean circulation during the last glacial – interglacial transition (Skinner et al., 2019; Telesiński et al., 2021). Outside of the Holocene, our $\Delta R$ values provide a minimum bound for calibration of radiocarbon ages, as also suggested in Heaton et al. (2023).

Although we provide a large expansion of spatial coverage around Greenland, large geographic areas remain unrepresented.

No data at all is available for the northern coast of Greenland bordering the Arctic Ocean, above 76 °N on East Greenland and 77 °N on West Greenland in the Nares Strait.



**Competing interests**

None of the authors have any competing interests.

**Acknowledgements**

Many thanks go to curators Mattias Forshage and particularly Anna Persson of the Swedish Museum of Natural History in Stockholm for providing access to the museum catalogues, the mollusk collections, and all the help with sampling. In Copenhagen, at the Zoological Museum (Natural History Museum of Denmark) we were welcomed by Tom Schiøtte, whose expertise of the Danish historical expeditions and mollusk collections was crucial for obtaining the samples needed for this study. Thanks to Alexis Geels and Mériadec Le Pabic for help with sampling, and photography of the catalogues and specimens

during museum visits. Finally, many thanks to Marie Kanstrup, technician at the Aarhus AMS Centre for the help with pre-treatment and analyses of radiocarbon measurements. This project was funded by the Independent Research Fund Denmark as project funding for CLAMS (Carbon Lag of Arctic Marginal Seas) to Christof Pearce (Grant 8021-00148B).

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
