# Peer review of "The marine reservoir age of Greenland coastal waters"

_Geochronology, 2023_

## Author Comment (AC1)

**Reply RC1: 'Comment on gchron-2023-7', Paula Reimer, 13 May 2023**

All reviewer text is in red. Author replies in black.

Marine radiocarbon calibration requires an estimate of the reservoir offset from the marine calibration curve (ΔR).  These estimates can be based on $^{14}$C measurements of pre-nuclear weapons testing, known age shells, independently dated coral, or contemporaneous marine and terrestrial samples. Until now the ΔR values for coastal Greenland have been sparse.  The authors have significantly enlarged the dataset of known age shell measurements from coastal Greenland and neighboring regions of the Arctic.  They have carefully selected samples from museum specimens to ensure the mollusks were collected alive.  The effects of sea ice cover, water depth and mollusk feeding habits were investigated and discussed. Regional averages were calculated for zones based on "prevailing currents and water masses" although most of the zones have overlapping values.  The authors also compared ΔR values for a limited number of samples stored in ethanol to dry samples.

Specific comments/questions:

Wet vs dry sample comparison:  This comparison is based on only 6 dry samples and 4 wet samples from one region (Suppl. Fig 2). This is a rather small dataset to reach the conclusion that dry samples are not reliably collected alive. It is difficult to tell which dry samples were used in the comparison but, of the 5 dry samples from Kaiser Frans Joseph Fjord, 4 were species with unknown feeding habits or deposit feeders.  It is well known, and also shown in this manuscript, that deposit feeders may incorporate older carbon from their environment. This comparison apparently forms the basis for one of the stated criteria for sample selection (line 412-413): 'Museum sample storage: As the exact age of samples from "dry" collections is possibly unknown, only samples with soft tissue present, stored in "wet" collections, should be used for ΔR evaluation'.  Samples stored in ethanol may be ideal to ensure live collection but this criterion would exclude many of the existing values in the literature.  In some cases, the museum documentation is unambiguous about live collection but there are also other indications of whether "dry" bivalves in collections were most likely collected live or shortly after death.  These include fragile mollusks that would have been abraded if transported to a beach as well as those with residual ligament, muscle or periostracum (O'Connor et al. 2010).  In addition, some species have colours that are light sensitive so would be bleached if not collected alive and stored in the dark (Angulo et al. 2007).

This is a good point, and we will make changes in the text to tone down this strict criterium for selecting museum samples. Where possible, "wet" samples are preferred,

but we will acknowledge that useful information can definitely be obtained from "dry" samples.

The study also makes use samples from water depths that would not be considered surface ocean in general. The low ΔR values for these samples provide a very interesting and useful observation for these locations which are 'characterized by convection and formation of North Atlantic Deep Water and Labrador Sea Water'. The authors advise that: 'When calibrating benthic dates from deeper sites one could therefore consider excluding extreme values obtained from surface ocean samples when making the choice of which reservoir correction to apply'. This seems like valid advice for these regions however it should be noted that surface ocean ΔR values are not generally applicable for benthic dates in other regions where deep water can be very depleted in $^{14}$C. Ideally one would have ΔR values from deep water samples to use for radiocarbon calibration of benthic samples but these are scarce in the literature.

Thank you for these nice comments. It is true that these deep "young" samples are probably more the exception than the rule, because of their location in areas of deep water formation. We will make changes to the text to make sure that this advice should not be applied in other areas where the ΔR of deeper waters is not known, or not showing this pattern of younger ages.

Also is there an explanation for the low ΔR values for relatively deep samples in NW Greenland zone 5? Is the West Greenland Current fed by Labrador Sea water?

This is an interesting observation. Indeed, in Figure 3 it is clear to see that multiple deeper sites (>500m) on the NW Greenland shelf have low ΔR values. We attribute this to the presence of young Atlantic Water at depth in the WGC, originating from the East Greenland Current mostly, but indeed Labrador Sea water can play a role here also. We will include this observation and discussion to the manuscript.

Technical comments:

If no reply is listed, we agree with all the below suggestions, and will make adjustments to the manuscript accordingly.

Line 18: " Marine20, the most recent radiocarbon calibration curve"  Insert "marine" ahead of radiocarbon.

Line 19 and 74: 'we introduce the term ΔR$_{13}$".  This term has been previously introduced in Heaton et al. 2023.  I would suggest replacing 'introduce' with 'use'

Line 51: 'to a lesser extent, injection of 14C-depleted CO2 from the burning of fossil fuels' Although this is a common perception and definitely true for reservoir ages relative to the atmosphere, for ΔR this is insignificant. ΔR is the difference between the marine radiocarbon age and the marine calibration curve which is modelled with input from the atmosphere so includes the Suess effect.

Line 54: 'tephrochronology (Pearce et al., 2017; Austin et al., 1995; Olsen et al., 2014), or paired marine/terrestrial dating' ΔR values may also be determined by U-Th dated coral (e.g. Hua et al. 2015).

Line 57: 'Several hundred different studies were made to study the local reservoir age'. Replace 'were' with 'have been'.

Line 127: 'the most commonly used value for the reservoir age correction (prior to publication of Marine20), ΔR = 0 14C years' Since ΔR without a subscribe is defined earlier as relative to Marine20 ,it would be better if this written here as 'Rxx = 0 14C were xx =04, 09 or 13.

Line 184: 'Wet samples were placed in a drying oven at 40 °C for several days' It would be worth stating that this is to remove any ethanol from the shell since contamination from the ethanol might be a concern.

Line 189: 'milliQ water' Trademark symbol needed

Line 239: 'where ΔRi and σi are the mean value and uncertainty of calculated local reservoir age offset'. Add 'of sample i' to clarify.

Line 243: 'Where the subscript w indicates that the uncertainty is calculate using the error each ΔRi' Change 'error' to 'uncertainty' and 'is calculate' to 'is calculated'

Line 368: 'no ΔR values higher than 50 years are found, and where ΔR values exceed 160 years,' ΔR values should be given as '14C yrs' rather than 'years'

Line 370: 'there is also a positive correlation between sea-ice cover and reservoir age'. Are the correlations significant?

Following your question, we have calculated the correlation and significance, and only for Zone 2 we found a significant positive correlation between sea ice concentration and ΔR. We will add this to the results description and the figure caption.

Line 429: 'these values remain only valid for the modern situation' Insert 'pre-bomb' before modern because the values would not be valid for post-bomb samples.

Fig. 1 caption: Need to define WGC, NFL, EGC.

Also. 'Areas of deep convection in the Labrador Sea and north of Iceland are colored yellow'. These look light green on top of the blue background - perhaps 'shaded light green' would be better

Fig 2.  Given the results, is there justification for separate zones for the Greenland coastal waters since ΔR values overlap?

It is correct that the values overlap, but our hypothesis was that the values would follow the different prevailing currents and water masses. We believe therefore that it is still valid to show these zones in Figure 2. In the discussion we mention that these regional averages are provided, but we encourage users to consult the full dataset before making decisions on which value to use. Other factors such as water depth can play an equally important role.

Suppl. Fig 2. Sample numbers on Suppl. Fig 2 would be helpful for comparison of species and feeding habits

This is a good idea. We will add the sample numbers to the markers on the map.

References:

Angulo, R. J., Reimer, P. J., De Souza, M. C., Scheel-Ybert, R., Tenório, M. C., Disaró, S. T. & Gaspar, M. D. 2007. A tentative determination of upwelling influence on the paleo-surficial marine water reservoir effect in southeastern Brazil. Radiocarbon, 49, 1-5.

Heaton, T. J., Bard, E., Bronk Ramsey, C., Butzin, M., Hatté, C., Hughen, K. A., Köhler, P. & Reimer, P. J. 2023. A response to community questions on the MARINE20 radiocarbon age calibration curve: marine reservoir ages and the calibration of 14c samples from the oceans. Radiocarbon, 65, 247-273.

Hua, Q., Webb, G. E., Zhao, J.-X., Nothdurft, L. D., Lybolt, M., Price, G. J. & Opdyke, B. N. 2015. Large variations in the Holocene marine radiocarbon reservoir effect reflect ocean circulation and climatic changes. Earth and Planetary Science Letters, 422, 33-44.

O'Connor, S., Ulm, S., Fallon, S. J., Barham, A. & Loch, I. 2010. Pre-bomb marine reservoir variability in the Kimberley region, Western Australia. Radiocarbon, 52, 1158-1165.

---

## Author Comment (AC2)

**Reply to RC2: 'Comment on gchron-2023-7', Anonymous Referee #2, 14 Jun 2023**

All reviewer text is in red. Author replies in black.

Pearce et al. present about 100 new marine radiocarbon ($^{14}$C) reservoir ages (MRA) of coastal and shelf waters around Greenland, Baffin Island, Newfoundland, and Iceland. The data result from $^{14}$C measurements on pre-bomb molluscs retrieved from museums. The MRA results are binned to seven regions and discussed with respect to the global Marine20 $^{14}$C calibration curve in terms of the regional MRA correction, $\Delta R_{20}$. The authors also discuss their $\Delta R_{20}$ results in the light of specific factors such as sample depth, sea ice cover and feeding habits.

The manuscript is well written, the presentation is clear, and the dataset is an important contribution to the MRA / ΔR data base. However, there are a few minor issues that should be addressed before publication in GChron (L = line):

L 30: The half-life of $^{14}$C has been slightly revised to 5700 years (e.g., Audi et al., 2003; Bé and Chechev, 2012; Kutschera, 2013)

Thank you for pointing this out. We will correct this and add a reference.

L 124 "marine mammals": "marine" should be removed

Agreed.

Figure 1:

 (i) Add a depth scale (such as in Fig. 1 by Pieńkowski et al. 2022)

Good point. Will add a scale for the bathymetry.

 (ii) "NFL", "WGC", and "EGC" should be also explained in the caption.

Agreed.

L 214-216 (and Figure 2): Is there a hard objective criterion to separate the three southernmost data points in East Greenland from region 7?

Yes. The reasoning here is that south of the Denmark Strait, there is more influence of Atlantic Water, caused by the southward bending Irminger Current (see also Fig. 1). These boundaries for Zone 6 are explained in the text in lines 218-220.

Figure 2: Explain "CS"

Cumberland Sound. All abbreviations in the Figure 2 caption will be explained.

L 353: Explain "mwd"

Meter water depth. This was explained in Line 272.

Figure 3: Would it make sense to indicate the positions of the outliers in the inserted map?

The outliers are actually included in the insert map, but we realize that their color (light grey) can be confused with land in the current color scheme. We will make sure that this is resolved in a revised version of figures 3 and 4.

L 376: Explain "mwd"

Meter water depth. This was explained in Line 272.

Figure 4:
 (i) As ΔR depends on the sea ice concentration, the coordinate axes should be swapped. The situation is different from Figure 3 where the independent variable (usually plotted along the horizontal axis) is depth (typically plotted in vertical direction).
We agree that the dependent variable should normally be on the y-axis, but as you already mentioned, this is not possible in Figure 3 because of depth being plotted vertically. Because figures 3 and 4 use the same styling and layout, we would prefer to keep ΔR on the x-axis for consistency. We believe that it is much easier for the reader to compare the figures if the data is presented in the same way.
 (ii) Can you quantify the trends, and are they significant? I wonder if the trends are still visible once the coordinate axes have been swapped.
The dotted lines plotted in Figure 4 are just to illustrate the general trends within each zone and the agreement with the theory of more sea ice resulting in older surface waters. We now have however calculated the trends and significance and found that a significant positive correlation between sea ice and ΔR is only present in Zone 2 (W Labrador Sea). We will make this clear in a revised version of the manuscript. As mentioned in the discussion, ΔR depends on several factors, and to get a real quantified relationship with sea ice, one should eliminate other variables such as water depth or mollusk species. Unfortunately, we don't have a large enough dataset to perform these analyses.
 (iii) Would it make sense to indicate the position of the outlier in the inserted map?
As for Figure 3, we will adjust the color palette, so the outliers are more visible

compared to land. The outlier is also referred to in the figure caption with the number reference to the supplementary table for more information.

**References:**

Audi, G., Bersillon, O., Blachot, J., and Wapstra, A. H.: The Nubase evaluation of nuclear and decay properties, Nuclear Physics A, 729, 3–128, https://doi.org/10.1016/j.nuclphysa.2003.11.001, 2003.

Bé, M.-M. and Chechev, V. P.: 14C - Comments on evaluation of decay data, Laboratoire National Henri Becquerel, Gif-sur-Yvette, http://www.lnhb.fr/nuclides/C-14_com.pdf, 2012.

Kutschera, W.: Applications of accelerator mass spectrometry, International Journal of Mass Spectrometry, 349–350, 203–218, https://doi.org/10.1016/j.ijms.2013.05.023, 2013.

---

## Author Comment (AC3)

**Reply CC1: 'Comment on gchron-2023-7', Elisabeth Michel, 23 Jun 2023**

All reviewer text is in red. Author replies in black.

The authors present new ¹⁴C reservoir ages for surface and deep waters of the North Atlantic and Nordic seas : Labrador sea, Baffin Bay and Iceland Sea, from shell museum collections. The shells have been collected from 1865 to 1931. They present a nice review of existing reservoir ages.

First, they compare the results from shells that were preserved in ethanol in museum collections and those who were dry samples. They found that the mean dry samples ¹⁴C reservoir age is much higher than the mean of ethanol preserved samples and argue that the dry samples might be dead since a long time when they were collected.

The authors propose regional ¹⁴C reservoir ages within 7 different geographic zones, considering both their new results and 14C reservoir ages from the Marine Reservoir Age Database (Reimer and Reimer 2001) considering only samples preserved in ethanol.

For the relevance of the results, the authors also consider the results of deposit feeders compared to suspension feeder.

For the interpretation of the regional ¹⁴C reservoir age they consider the depth of collection of the different samples and shortly discuss the impact of ocean circulation and sea ice.

This paper is mainly a data paper, the discussion of the result is rather short and do not discuss in depth the different factors that could impact their regional ¹⁴C reservoir age.

This is partly true, but with little available data, one can also argue against going into too much depth with the discussion of different influences. We believe that we have covered the main influences on the regional reservoir ages in broad terms in the discussion, but indeed there are several other factors that could be added and combined in our investigation. The limitation here becomes the number of data points available to investigate the combination of different factors. One could investigate e.g. the influence of mollusc feeding habit per region, but in most cases the number of samples is too low to infer any significant relationships. We have therefore included the full dataset in the supplementary information, where we provide more details than are

included in the discussion. This allows individual users or detailed follow-up studies to use the complete available data to provide regional reservoir age estimates.

Following are some detailed comments and also some ideas for a more complete discussion concerning the regional results.

Considering dry samples, I wonder if there is any evidence on the shell, muscle marks or the like, to tell whether the specimen was collected alive or could have been dead for a long time.

As you suggested, it is possible to look for such evidence on dry samples, but in this study, we unfortunately did not investigate this. Another reviewer also suggested that we tone down our recommendation of simply excluding the dry samples. Based on both of your comments, we will suggest that "wet" samples are preferred where possible, but "dry" samples can be included if carefully examined for signs that would indicate if they were recently alive. We will include this in the revised manuscript and include some references.

For the deposit and suspension feeders, the authors should compare the results zone by zone as they indicated that the ΔR was very different from one zone to another. They could also check the dispersion for species for which the feeding habit is unknown. It would be better to discuss first the aspect linked to the mollusk : dry and ethanol preserved samples, feeding habitat and after all the physical parameters: sea ice, depth and circulation.

This is a great suggestion, but unfortunately there is not enough available data to investigate this in detail. There are approximately 4 times more suspension feeders as deposit feeders in the dataset (Figure 5), and per zone the differences between the groups are not significant. Zones 3 and 4 have the highest ΔR values, but there is no clear difference between the different feeding habits. In Zone 4 (Nares Strait), the values for the few deposit feeders fall right in the middle of those of suspension feeders, while in NW Baffin (Zone 3), the deposit feeders represent both the highest and the lowest ΔR values. The species with unknown feeding preferences are unfortunately represented in even lower numbers.

One question that is not addressed, do the author have an idea of the mean lifetime of the different mollusk?

This was indeed not included, thanks for pointing this out. The lifespan of mollusks is extremely variable between different species and individual specimens and can range from years to decades, to even centuries. To avoid this issue, we made sure to always sample material from the outermost part of the shell, i.e., the carbonate of youngest

age. This was stated in the methods section, but we will expand this a bit to include more explanation and a reference.

It seems that the authors choose to include only ¹⁴C ΔR measured on molluks. I wonder why they do not compare their results with 14C measurements made directly on DIC of sea water in the early fifties like for example Fonselius and Östlund, 1959 Tellus.

The aim of this study and the dataset is to improve calibrations of radiocarbon datings on marine sediment cores for use in paleoceanography. This is why we restrict ourselves to mollusks since marine carbonate fossils are the main source for radiocarbon dates in these sequences. We hope that our dataset and analyses can be of use by oceanographers as well, but we believe this is outside the focus of our study.

What is the most impressive is the dispersion of the ¹⁴ΔR data within some of the geographic zones. The authors discuss the impact of sea-ice checking if a relationship exist between the annual average sea ice concentration of a sample location and its ¹⁴C reservoir age (fig. 4). The regressions and their statistics for the different geographic zones are necessary if the authors want to demonstrate that the regional relationships are significant. Furthermore during formation of sea ice the carbon sink in the ocean might be effective thus the impact of non-perennial sea-ice is not obvious.

Thank you for this suggestion, which also came up in two other reviews. We have done some significance tests and found that only for Zone 2 we have a significant positive correlation between sea ice and ΔR. For the other regions, the correlations are either too weak, or there are too few datapoints for detecting a significant trend. Zone 2 not only has a large latitudinal range, but it also includes the most samples of any of the zones. We will add this information about significance of the trends to the text.

The authors argue that Heaton et al., 2020, explain that the Marine20 does not apply to the polar regions because of sea ice. Heaton et al, 2020 is as much about ocean circulation as it is about sea ice.

Heaton 2020 specifically discuss the influence of sea ice on the reservoir age but they indeed also mention ocean circulation.  We will add this to our introduction.

The role of Ocean Circulation could be considered considering fluxes along the different straits. Furthermore the influence of Atlantic and Artic water masses might changes with time, for example linked to North Atlantic Oscillation. Thus a time evolution of ¹⁴C ΔR within the geographical zones could be also discussed and might explain partly the large dispersion of the results?

This is true, but again we are limited by the availability of data. To study proper time series, we would need temporally spaced ΔR values which ideally would come from single specimens, or the same species from similar localities, including water depth. Right now, when the data is plotted by calendar year (see figure below), it is highly regionally clustered around certain years in which the major expeditions took place. Although certain regions were visited during different years, the samples are then not from the same locations, or of the same species.

[Figure]

ΔR could be also influence by continental waters with old ¹⁴C DIC coming from under the ice like in the Ross Sea (Mikucki et al., 2009). This point is not discussed.

This is a good point and indeed it is something we have not mentioned. In the introduction we do list continental runoff as a possible influence, but not specifically the input of under-ice pre-age waters. We will include this in the discussion of the spatial variability of the ΔR values. Based on our data however, we don't have the suitable samples to investigate this process.

Figures: even if the projections does not make it easy and they will not be regularly spaced, it would be nice to have some latitudinal and longitudinal tics on the borders of figures 1, 2 and suppl. Fig.1.

We left those out to avoid cluttering the main figures even more, but we see your point. We will add a lat-lon grid to the map of Supplementary Figure 1 which shows the positions of all different samples.

---

## Author Comment (AC4)

**Reply to RC3: 'Comment on gchron-2023-7', Matt O'Regan, 29 Jun 2023**

All reviewer text is in red. Author replies in black.

This is a very nicely written paper presenting 92 new radiocarbon dates on pre-bomb mollusks collected from around Greenland (with the exception of its northern Arctic Ocean margin). In addition to the utility of these new dates for constraining regional reservoir corrections, I think the manuscript is timely in presenting a nice practical discussion (and examples) on the need to update reservoir corrections when using the new Marine20 calibration curve.

The comparisons of dR with water depth and sea ice coverage are interesting in highlighting patterns, although somewhat inconclusive in identifying a cause/explanation for the variability. I do not think this limits the scientific contribution made by the paper, and certainly sets the stage for future work needed to understand this variability. I believe this would require a considerable amount of work, and could potentially start with moving away from water depth and looking at the variability in Temperature-Salinity space to see if ages cluster in specific water masses. However, I do not think this is a necessary addition to this work, which very well suited for publication in Geochronology in its current form.

Thank you very much for these comments and suggestions. The idea of mapping out the results over water masses is very interesting and, as you suggest, definitely worth looking into for follow-up studies.

I do feel one aspect the paper is missing is a discussion on the limited, but rather informative data on Holocene dR values from the central Arctic Ocean. Specifically the inferred differences between the age of Pacific and Atlantic waters that are found in the interior Arctic, and should be impacting the age of surface waters(?) in northern Baffin Bay. For example, West et al (2022), *Geochemistry, Geophysics, Geosystems* (doi: 10.1029/2021GC010187) used tephra from the Aniakchak eruption circa 3.6 ka in two cores from the Chukchi Sea - one at 50 m depth (Pacific water) and one at 120 m depth (likely Atlantic water) - to show that the dR (using Marine 20) for benthic foraminifera and mollusks at these sites was about 330 years for Pacific waters and 205 years for Atlantic waters. These seem to be somewhat consistent with the larger dR values in sections 3 and 4 from Northern Baffin Bay. It would be nice to see some discussion about the influence of Arctic outflow and the water masses involved on the dR values in Northern Baffin Bay. Currently these are described simply as 'outflow' from the Arctic, which could easily be expanded to detail the role and age of Pacific and Atlantic waters in this outflow.

This is an excellent point, and we will make sure to include this in a revised version of the manuscript.

Overall, I feel this is a great contribution that will provide significant support to future paleoceanographic work around Greenland.